# Characteristics of Immunoglobulin M Type Antibodies of Different Origins from the Immunologic and Clinical Viewpoints and Their Application in Controlling Antibody-Mediated Allograft Rejection

**DOI:** 10.3390/pathogens10010004

**Published:** 2020-12-23

**Authors:** Yoshiko Matsuda, Takahisa Hiramitsu, Xiao-kang Li, Takeshi Watanabe

**Affiliations:** 1Division of Transplant Immunology, National Research Institute for Child Health and Development, Tokyo 157-8535, Japan; ri-k@ncchd.go.jp; 2Department of Advanced Technology for Transplantation, Osaka University Graduate School of Medicine, Osaka 565-0871, Japan; 3Department of Transplant and Endocrine Surgery, Nagoya Daini Red Cross-Hospital, Aichi 466-8650, Japan; thira@nagoya2.jrc.or.jp; 4Laboratory of Immunology, Institute for Frontier Life and Medical Sciences, Kyoto University, Kyoto 606-8507, Japan; tthta_w@icloud.com

**Keywords:** Antibody-mediated allograft rejection (AMR), donor-specific antihuman leukocyte antigen antibodies (DSAs), IgM antibodies, innate B cell, conventional B cell

## Abstract

Antibody-mediated allograft rejection (AMR) hinders patient prognosis after organ transplantation. Current studies concerning AMR have mainly focused on the diagnostic value of immunoglobulin G (IgG)-type donor-specific antihuman leukocyte antigen antibodies (DSAs), primarily because of their antigen specificity, whereas the clinical significance of immunoglobulin M (IgM)-type DSAs has not been thoroughly investigated in the context of organ transplantation because of their nonspecificity against antigens. Although consensus regarding the clinical significance and role of IgM antibodies is not clear, as discussed in this review, recent findings strongly suggest that they also have a huge potential in novel diagnostic as well as therapeutic application for the prevention of AMR. Most serum IgM antibodies are known to comprise natural antibodies with low affinity toward antigens, and this is derived from B-1 cells (innate B cells). However, some of the serum IgM-type antibodies reportedly also produced by B-2 cells (conventional B cells). The latter are known to have a high affinity for donor-specific antigens. In this review, we initially discuss how IgM-type antibodies of different origins participate in the pathology of various diseases, directly or through cell surface receptors, complement activation, or cytokine production. Then, we discuss the clinical applicability of B-1 and B-2 cell-derived IgM-type antibodies for controlling AMR with reference to the involvement of IgM antibodies in various pathological conditions.

## 1. Introduction

Many reports have described the close relationship between donor-specific antihuman leukocyte antigen antibodies (DSAs) and the development of antibody-mediated allograft rejection (AMR) and the difficulties in eliminating long-lived bone marrow plasma cells (PCs), which are known to produce IgG-type DSAs [1,2]. In transplant recipients, the appearance of IgG antibodies against donor-specific human leukocyte antigen (HLA) in serum has been used as a diagnostic sign of AMR from the viewpoint of their antigen specificity. However, the pathology of AMR has proceeded irreversibly by the time these antibodies are detectable in serum [3]. Conversely, the IgM antibody is usually inactivated by the reducing agent dithiothreitol because it hampers the detection of antigen-specific IgG antibodies using multiplex bead-based flow cytometry assays through direct competition between IgM and IgG antibodies, steric hindrance, or complement activation by the IgM antibody pentamer [4].

Recently, some reports claimed the involvement of IgM antibodies in the activation of humoral immunity against donor-specific HLA [5,6], but other studies suggested that these antibodies are not clinically involved in the development of AMR [7]. Thus, no consensus has been reached so far regarding the role of IgM antibodies against donor-specific HLA in the context of AMR development.

Unlike IgG antibodies, most serum IgM antibodies consist of B-1 cell (innate-like B cell)-derived natural IgM antibodies rather than B-2 cell (conventional B cell)-derived IgM antibodies [1,8]. Thus, it is possible that the antigen polyreactivity of natural IgM antibodies hinders the detection of B-2 cell-derived IgM antibodies using immunobead assays, and may cause false-positive reactions. In addition, the clinical significance of B-2 cell-derived IgM antibodies in AMR may not be accurately evaluated if they are masked by B-1 cell-derived natural IgM antibodies [4,9,10,11,12]. Therefore, such a difference between IgM and IgG antibodies may contribute to the inconsistent roles of IgM antibodies in the context of AMR development.

Additionally, antibodies against ABO blood group antigens, which are representative of IgM and IgG antibodies, induce the development of AMR, causing transplanted graft failure in ABO-incompatible liver transplant recipients [13,14,15].

Concerning the association of the type of transplanted organs with AMR development, AMR has not been overcome in ABO-incompatible liver transplantation from brain-dead donors, but it is mostly controlled in ABO-incompatible liver transplantation from living donors, and in ABO-incompatible kidney transplantation, through improvements in immunosuppression protocols [13,14,16,17,18,19,20,21,22]. 

Meanwhile, accommodation is also induced in both kidney and liver ABO-incompatible transplantation, thereby protecting the transplanted organ from the recipient humoral immune system [20,23,24,25]. However, the detailed mechanism by which blood group antibodies participate in accommodation induction is not fully understood.

Additionally, IgM antibodies are involved in the activation of complement and immune responses via the FC mu receptor (FcµR) expressed on immunocompetent cells, as discussed in Section 3.3 and Section 3.4.

Therefore, this review discusses the direct effects or the involvement of humoral immune response in transplanted grafts via the aforementioned pathway of IgM antibodies, considering the origin, corresponding antigen, differences in the target organs, and other conditions in order to clarify the clinical significance of IgM antibodies in the field of transplantation.

## 2. Molecular Pathophysiology

### 2.1. B-1 Cells

B cells that produce IgM antibodies are classified into two lineages: B-1 and B-2 cells. Most serum IgM antibodies consist of natural IgM antibodies derived from B-1 cells. These antibodies participate in the neutralization of harmful pathogens and activation of complement and adaptive immunity. B-1 cells also induce CD4^+^ T cell differentiation into regulatory T cells, which play an important role in maintaining peripheral tolerance [26]. B-1 cells express macrophage-1 antigens and exhibit low CD45RA expression. Most of these cells exist in the body cavity and intestinal tract [27,28] and account for 15% of the peripheral blood B-lymphocyte population [8].

Environmental factors such as self-antigens and thymic stromal lymphopoietin (TSLP), which are produced by epithelial cells and fibroblasts, and signals from interleukin (IL)-5 receptors play important roles in maintaining the survival of these cells [29,30]. Cells with low reactivity for self-antigens are selected and those with high reactivity for self-antigens are excluded through apoptosis during differentiation into IgM antibody-producing cells [31,32]. Additionally, B-1 cell-derived IgM antibodies with at most 10 antigen-binding sites, can react with multiple epitopes such as phospholipids, oxidized lipids, glycolipids and glycoproteins expressed on thymus-independent antigen with high avidity in a polyclonal manner [33,34]. Thus, these IgM antibodies can react with the epitope expressed on self-antigens, and some of these antibodies are capable of reacting with the epitopes expressed on foreign antigens [27,28,31,35,36,37]. Additionally, B-1 cells also produce IL-10, which is an anti-inflammatory cytokine and regulates the function of B-1 cell and suppresses B-1 cell proliferation in response to B cell receptor (BCR) ligation and downregulates the expression of proinflammatory cytokines, helper T (Th) cell cytokines, major histocompatibility complex (MHC) class II antigens, and costimulatory molecules on macrophages [36,38,39,40,41], which results in the downregulation of excessive immune responses to self-antigens [42].

In addition, signals from Toll-like receptor (TLR) ligands and CD40, which are highly expressed on B-1 cells during infection, activate B-1 cells in an anergic state [39,43,44]. Signals from TLRs induce the production of the proinflammatory cytokine IL-6, which promotes the differentiation of B-1 cells into antibody-producing cells in the presence of IL-21 secreted from Th cells.

B-1 cells are further classified into B-1a (CD5^+^) and B-1b cells (CD5^−^) [39,40,45]. B-1a cells develop from fetal liver stem cells. Signals from CD5 receptors downregulate the expression of B cell receptors (BCRs) on B-1a cells, which exhibit cross-reactivity with self-antigens. Excessive cross-linking of BCRs with self-antigens is known to induce apoptosis in these cells. Thus, CD5 expressed on B-1a cells limits the production of excessive autoantibodies by B-1a cells and the negative regulation of BCR signaling by CD5 expression plays a vital role in preventing autoimmune disease development. 

B-1a cells can, reportedly, spontaneously secrete a germline-like, polyreactive natural antibody (IgM), which acts as the first line of defense by neutralizing a wide range of pathogens. B-1a cells are also known to produce several immunomodulating molecules, either spontaneously or when stimulated, which attenuate infectious and inflammatory diseases, atherosclerosis, inflammatory bowel disease, autoimmunity, obesity and diabetes mellitus. B-1a cells play a remarkably protective role against acute lung injury induced by sepsis by controlling exaggerated inflammation and infiltration of neutrophils in lungs [46] These cells also reportedly produce IL-10, which suppresses the proinflammatory response and contributes to the resolution of inflammation-induced injury. These reports indicate that B-1a cells could play a crucial role in immunoprotection against inflammation, tissue injury and immunological attacks, either by secreting natural IgM antibodies or IL-10 as well as other immunoregulatory molecules [47,48,49,50,51,52,53,54,55]. 

Meanwhile, B1-b cells have the capacity for self-renewal as in the case of B-1a cells. However, B-1b cells do not express CD5, and they develop from precursors in the fetal liver and adult bone marrow, unlike B-1a cells which express CD5 and develop in the fetal liver [56]. Thus, BCR signaling in B-1b cells may be negatively regulated by mechanisms other than CD5 expression, and BCR signaling, rather, positively regulates B-1b cell development [36,57]. In addition to this, B1-b cell-derived antibody-producing cells produce antigen-specific IgM antibodies and can undergo immunoglobulin class switching to IgG3- or IgA-expressing cells by cross-linking BCR expressed in these cells [58,59,60,61,62,63]. 

Reports have indicated that B1-b cells have an antigen-presenting function, an anti-inflammatory effect and phagocytic activity, and in vitro findings indicate that the functions acquired by B-1-b cells differ depending on the type and combination of the substrate attached to the B1-b cells while cultured in the following manner.

Soluble polysaccharides (PS) induce the differentiation of IL-10-expressing B1-b cells, which have an anti-inflammatory effect in the abdominal artery via IL-10 production, even under in vivo conditions. Bacterial invasion induces B1-b cells to differentiate into MHC class 2-positive antigen-presenting cells. The combination of *P. acnes*, a Gram-positive bacillus, and PS induces differentiation into phagocytes with higher phagocytic activity. Therefore, the possibility of B1-b cell differentiation into cells with these diverse functions plays an important role as a bridge between innate and adaptive immunity [27,41,64,65,66] (Figure 1). 

### 2.2. B-2 Cells

Conventional B cells (CD5^−^ B cells, also known as B-2 cells) are delivered from the bone marrow. Most of these cells exist in secondary lymphoid tissue and peripheral blood. They account for 45% of peripheral blood B-lymphocytes [8]. These CD27^−^IgD^+^ naïve B cells undergo clonal expansion following exposure to antigens and stimulation by activated CD4^+^ Th cells presented with antigen by antigen-presenting cells via the major histocompatibility complex (MHC) Ⅱ/T cell receptor (TCR) interaction during initial infection. These CD27^+^IgD^−^activated B cells undergo isotype class switching to IgG^+^, IgA^+^ and IgE^+^ cells, and some cells differentiate into IgM^+^ cells. Subsequently, these cells form the germinal center within the secondary lymphoid tissue for further affinity maturation [45,67].

In the germinal centers, the antigen specificity of IgM^+^IgD^−^ B cells, as well as switched B cells, highly diversifies and maturates via gene conversion and hypersomatic mutation. B cells with low affinity for foreign antigens, or high affinity for self-antigens, are excluded through apoptosis as a negative selection mechanism [68,69].

Various cytokines play an important role in germinal center reactions. It has been reported that the addition of IL-21, CpG oligodeoxynucleotides (CpG-ODN) and the CD40 ligand could sustain the growth and survival of memory B cells (mBCs) into antibody-producing cells in an in vitro assay system [70]. The method by which each cytokine participates in the growth and survival of mBCs has been reported. Specifically, IL-21 produced by CD4^+^ Th cells directly maintains the differentiation and proliferation of B cells [71].

CpG-ODN is a ligand for TLR9, which can promote IL-6 production by B cells upon stimulation by CpG-ODN and activate B cell growth and survival in a T cell-independent manner. TLR9 also increases the population of CD4^+^ Th cells and promotes interactions between follicular Th cells and B cells [72,73,74].

CD40 expressed on the surface of B cells promotes B cell differentiation and proliferation as well as induction of the class switch reaction (CSR) and somatic hypermutation (SHM) by interacting with the CD40 ligand expressed on the surface of CD4^+^ Th cells [75].

In addition, it has been reported that IL-4, IFNγ, IL-10, and other cytokines produced by CD4^+^ Th cells, also play important roles in B cell differentiation, CSR, SHM and CD4^+^ Th cell differentiation and activation in germinal centers [76,77,78,79,80]. Meanwhile, IFNγ may be required for the production of IgM mBCs corresponding porins in bacteria in the germinal center, because it is accompanied by the expansion of IFNγ-producing CD4^+^ Th cells [81].

The surviving cells, after the germinal center reaction subsequently differentiate into either CD27^+^IgM^+^IgD^−^mBCs or long-lived PCs, which continue producing IgM antibodies with high affinity for foreign antigens in the bone marrow for an extended period of time (Figure 2).

It has also been reported that signaling via CD40 promotes IL-9 receptor expression on the surface of mBCs, and IL-9 promotes differentiation mBCs into PCs through the IL-9 receptor [1,82].

In germinal centers, these cells differentiate with diversifying affinity into PCs, which produce IgM antibodies with a high affinity for T cell-dependent antigen.

Simultaneously, T cells are activated by antigen-presenting cells (APCs), and the activated T cells (follicular helper T cells; T_FH_) support B cell differentiation in germinal centers. Class II MHC expressed on the surface of antigen-presenting cells (APCs) presents antigens to T cell receptors and activates the T cells [83]. B7 family costimulator molecules (B7-1, B7-2) bind to the CD28 molecule and CTLA-4 on the surface of T cells and give an inhibitory signal on antigen-activated T cells. B7H-1 binds programmed death-ligand 1 (PD-L1) found on activated T cells and suppresses T cell activation through PDL1/PD-1 interaction.

B7H-2 binds ICOS found on CD4/CD8 T cells, and enhances secretion of IFN-γ and IL-10, which sustain immune balance [84,85,86,87,88].

Meanwhile, the CD40 ligand expressed on the surface of activated T cells binds to CD40 expressed on the surface of B cells and supports the B cell adaptive immune responses, including germinal center formation [69].

## 3. The Role of IgM Antibodies in Pathology

### 3.1. B-1 Cell-Derived Natural IgM Antibodies

The incidence of arthritis, cardiovascular disease and systemic lupus erythematosus (SLE) is extremely low in the presence of IgM class natural autoantibodies (NAAs), indicating that serum IgM class NAAs may protect self-antigens from immune attack or participate as negative regulators of inflammatory reactions during autoimmune disease.

Moreover, serum IgM antibodies inhibit the pathological function of IgG autoantibodies via several strategies, including the neutralization of IgG autoantibodies via anti-idiotypic activity, competitive inhibition of IgG autoantibody binding to antigens, inhibition of functional activity by binding to the antigen-binding fragment region of IgG NAAs and suppression of IgG autoantibody production through FcµR on the B cell surface [89].

Natural IgM antibodies have several protective roles, including the suppression of autoimmune reactions and inflammation in target organs. These protective effects by B-1 cells may be mediated by increasing the production of the anti-inflammatory cytokine IL-10, which suppresses the production of proinflammatory cytokines such as IL-6 and tumor necrosis factor-α in the Th17 response [38,39,49,90]. In addition, polyclonal IgM antibodies may promote the phagocytosis of apoptotic cells through activation of the classical pathway of complement, thereby alleviating pathological conditions through the elimination of these cells, or these IgM antibodies may directly participate in masking antigens and removing harmful molecules [31,35,69,90,91].

In a model of renal ischemia-reperfusion injury, the tissue-protective effects of IgM antibodies during the innate inflammatory response has been reported. Additionally, natural IgM antibodies recognize and block danger-associated molecular patterns (DAMPs). Furthermore, complement activation induced by IgM antibodies assists in the regeneration of damaged hepatocytes in a liver ischemia model [92].

Therefore, many reports have indicated that serum IgM antibodies, which consist mostly of natural IgM antibodies, play crucial roles in protecting target tissues from immune attack, and these antibodies may also be involved in repairing of damaged tissues.

### 3.2. The Role of B-2 Cell-Derived IgM Antibodies in Pathology

In order to clarify the involvement of B-2 cell-derived IgM antibodies in pathology, it may be useful to culture peripheral blood mononuclear cells (PBMCs) with a significantly higher number of B-2 cells and under the suitable culture conditions for the growth and survival of B-2 cells. Moreover, it is necessary to exclude the influence of naïve B cell-derived natural IgM antibodies because of the frequent presence of naïve B cells in PBMC populations [29,93]. A particularly useful method for the selective analysis of B-2 cell-derived IgM antibodies from culture supernatant is the reported observation that the period required for antigen-specific B cells to differentiate into antibody-producing cells is shorter than that for naïve B cells [69,94].

Several researchers have detected HLA antigen-specific mBC differentiation by using in vitro culture systems. Orio et al. stimulated the differentiation of IgG-positive mBCs in PBMCs into antibody-producing cells using recombinant human IL (Mabtech, Nacka Strand, Sweden) and the TLR7/8 agonist R 848 [95]. TLR7 is required for the development of germinal centers and the CD4^+^ Th cell response and TLR8 control TLR7 function [96,97]. They evaluated the frequency of DSA-specific mBCs in the PBMC pool/8via enzyme-linked immunosorbent assay and reported its utility for assessing the activation of the humoral immune response against donor-specific HLA, and for predicting AMR development.

Meanwhile, in order to differentiate mBCs into antibody-producing cells, Han et al. cultured CD19-positive B cells derived from kidney transplant recipients on CD40 ligand-expressing feeder cells (EL4-B5) together with growth factors and cytokines secreted by activated T cells in vitro. They compared IgG and IgM DSA in serum and culture supernatant to examine the accuracy of antigen-specific IgG and IgM antibody detection systems [98].

In addition, the possible clinical application for the control of AMR by detecting IgM^+^ HLA-specific mBC differentiation has also been reported. Specifically, the levels of humoral immune response activation can be more accurately evaluated by simultaneously analyzing DSA-specific IgG^+^ and IgM^+^ mBC differentiation, and the detection of DSA-specific IgM^+^ mBC- differentiation is useful as an early diagnostic method for AMR that enables early preventative therapeutic intervention [94,99].

### 3.3. Involvement of IgM Antibodies in Complement Activation

Natural IgM antibodies form an immune complex (IC) with an antigen and activate complement, and the IgM antibodies bind to the surface of apoptotic cells or macrophages through complement receptors. This enables the binding of C1q to the μ constant region, and the C1q complex activates the classical complement pathway by binding to the C1q receptor, subsequently initiating innate responses for preventing certain bacterial infections, neutralizing virions, and clearing dying cells in collaboration with activated C1q. Thus, complement activation also plays an important role when natural IgM antibodies defend against pathogens [90].

To enhance the antibody response, IgM antibodies bind to the antigen and activate C1q, leading to complement deposition on the antigen. The antigen-complement complex subsequently cross-links with BCR, and then the complex of the complement receptor 2 (CR2)/CD19/target of the antiproliferative antibody-1 (TAPA-1) coreceptor is formed on B cells, which results in lowering the threshold for B cell activation.

Alternatively, marginal zone B cells bind to IgM–antigen complexes via FcµRs, and the IgM–antigen complexes are trapped in the marginal zone in mice lacking C3 and CR1/2, permitting their movement toward follicular dendritic cells (DCs). Additionally, CR1/2 on follicular DCs facilitates the migration of complement-coated antigen toward follicular DCs. This transfer of the antigen to follicular DCs plays a vital role in the induction of an optimal antibody response. Thus, the antibody response is activated depending on the complement activity [100].

### 3.4. Involvement of IgM Antibodies in Immune Responses via the IgM Receptor (FcµR)

FcµR, to which IgM binds through its Fc portion, is expressed on the surface of B cells, regulatory T cells, NK cells, granulocytes, macrophages and DCs. Additionally, FcµR is also expressed in the transGolgi network, and it suppresses IgM-BCR expression on the B cell surface or regulates the function of T cells and DCs [101]. FcµR expressed in the transGolgi network restricts the transport of IgM-BCR to the B cell surface. This type of B cell subset is also associated with the degree of FcµR expression on B cells. FcµR is also expressed on immature B cells in bone marrow, in which it recognizes self-antigen and interacts with IgM-BCR more strongly than FcµR expressed on mature B cells. This interaction promotes the elimination/anergy of autoreactive B cells in bone marrow [31,102]. Pentameric IgM antibodies bind to the antigen and form a large IgM antibody–antigen complex. The complex regulates the survival and growth of B cells by cross-linking BCR with the complement receptor CD21 or FcµR. By contrast, monomeric IgM antibodies bind to B cells via FcµR and interact with antigen-bound BCR; thus, FcµR supports B cell growth and survival. Therefore, the B cell subset is also related to the involvement of FcµR in B cell survival and growth [34,101,102,103].

Pathologically, the FcµR-deficient mouse model displays increased levels of natural IgM antibodies in serum, spontaneous B cell activation, germinal center formation and autoreactive IgG antibody production. Thus, FcµR is responsible for regulating the B-1/B-2 cell ratio, maintaining the homeostasis of B-2 cells, controlling the production of harmful autoreactive IgG antibodies and activating mBCs. Therefore, this receptor is likely associated with the development of autoimmune diseases, chronic lymphocytic leukemia and other conditions [34]. Furthermore, FcµR expression is increased on the surface of macrophages after TLR4 stimulation and it negatively regulates the natural antibody-mediated immune response to T cell-independent antigens [90].

## 4. Significance of IgM Antibodies in Various Disease States

### 4.1. Infection

During infection and exposure to foreign antigens, IgM antibodies represent the first immunoglobulin produced and act as the first adaptive defense to protect the body. Therefore, the detection of serum IgM antibody is a potential method for early diagnosis. Because of false-positive reactions, other tests should be performed for confirmation [104]. For example, by evaluating both IgM and IgG antibody production (IgM–IgG antibody combined screening) as a diagnostic method for viral infection, specificity can be increased with higher sensitivity. Additionally, this test reveals whether the current infection is an initial infection or reinfection. For example, IgM positivity and IgG negativity indicate a recent primary infection, whereas IgM negativity and IgG positivity suggest subsequent reinfection [105,106].

### 4.2. Autoimmunity

Serum IgM antibodies against double-stranded DNA are more likely to be detected during inactive SLE than during active SLE. Serum IgM antibodies are detected more often in other sclerodermas, collagen diseases, chronic hepatitis and infectious mononucleosis than in SLE. Furthermore, although their specificity is low, these antibodies are associated with pathological conditions, and their levels tend to be lower during the active phase in follow-up observations in patients with SLE [107].

In rheumatism, IgM rheumatoid factor (RF) levels are also commonly measured. Although IgG RF is less frequently present than IgM RF, it has been reported to be more strongly correlated with the activity of extraarticular lesions [108]. Thus, clinical evaluation of both IgG and IgM autoantibodies is useful for the detailed evaluation of pathological conditions in the field of autoimmunity.

### 4.3. Cancer

In breast cancer, the detection of autoantibodies against tumor-associated antigens is expected to be useful for early diagnosis and prognosis, and IgG antibody-like molecules have been proposed as indices of therapeutic efficacy. However, IgG antibodies are usually subject to immunoregulation, and they reflect the suppression of the immune response against tumors. Conversely, natural IgM antibodies contribute to tumor elimination by directly presenting tumor antigen to NK cells and DCs. Natural IgM antibodies perform neutralization by directly binding to tumor cells, and they activate B cells via T cell activation. Therefore, the detection of IgM antibodies is expected to be useful as an early diagnostic method in combination with imaging analyses [109].

### 4.4. Clinical Significance of IgM-Based Therapies against Inflammation and Infection

Normal IVIG contains IgG antibodies at levels of 95% or more, but IgM antibody-enriched IVIG contains IgM antibodies at a much higher percentage, such as 90% [30,110,111]. The therapeutic effect of polyclonal IgM antibody infusion has been reported in clinical practice, and polyclonal IgM antibody-enriched IVIG has anti-inflammatory effects and suppresses complement activation and the production of IgG autoantibodies [30,110,111,112].

In a mouse model of artificially induced sepsis in the abdominal cavity, the injection of IgM antibody-enriched IVIG reduced lethality by neutralizing toxins together with erythropoietin [113]. It further enhanced the functions of immunocompetent cells, bacterial aggregation and opsonin activity. IgM antibody-enriched IVIG suppressed lung and small intestinal injury in a prior study [114,115]. Additionally, IgM-enriched IVIG effectively suppresses lymphocyte proliferation stimulated by phytohemagglutinin and the mixed lymphocyte reaction. Therefore, it can be expected to alleviate the pathology of autoimmune diseases by strongly suppressing T cell proliferation [111].

Infusion of the extracellular domain of the IgM receptor (FcµR) or anti-FcµR monoclonal antibody effectively suppresses inflammation and infection. Regulating signals from FcµR may be more important in controlling autoimmune and inflammatory diseases. Further elucidation of the mechanism by which serum IgM antibody participates in pathophysiology via FcµR is merited for its therapeutic application [103].

Thus, reports have indicated that IgM antibody may be useful for evaluating pathological conditions. Its clinical potential as a therapeutic method has been reported to include the control of inflammation and protection of target organs from immune attack.

## 5. The Role of IgM Antibodies against Donor-Specific HLA in the Field of Transplantation

### 5.1. Conventional Policy and Problems

Current studies have focused on the diagnostic value of IgG antibodies against HLA expressed in donor-derived vascular endothelial cells. However, bone marrow PCs that produce these antibodies survive independently of T cells and maintain reduced CD20 antigen expression on their surface [116,117]. Consequently, immunosuppressive therapy chronically targeting T cells, and monoclonal anti-CD20 antibody administration targeting CD20 antigen-positive B cells, are insufficient for controlling AMR. Desensitization therapy, which has serious side effects, is needed to control AMR [68,118]. Additionally, IgG DSA produced in serum is not detected in the early stage of production because it is absorbed by the transplanted grafts, and it is not expected to be useful as an early diagnostic marker that enables early therapeutic intervention. Therefore, the Transplantation Society does not recommend routine antibody monitoring beyond the first year in all transplant recipients, except in the following cases: (1) the efficacy of immunosuppression changes, (2) nonadherence is suspected, (3) graft dysfunction occurs, or (4) the patient is transferred to a remote outside center [119]. Therefore, an early diagnostic method for AMR and a less invasive control method are urgently needed.

### 5.2. Effect of IgM DSA on Transplanted Organ

Some reports indicated the involvement of IgM DSA in the activation of the immune responses against transplanted organs. In kidney transplant recipients, the production of both IgG3 and IgM antibodies against donor-specific HLA is persistently involved in the activation of the humoral immune response against the transplanted graft, and IgM DSA detection in serum is associated with transplanted graft survival [5]. In lung transplants, IgM DSA detection in serum was associated with graft survival [120], and high serum IgM DSA levels or the detection of the IgM antibody against class 1 HLA was associated with the development of chronic rejection and the development of bronchiolitis obliterans syndrome [121]. In heart transplants, IgM DSA was associated with transplanted graft survival [120]. Additionally, IgM DSA may be useful as a biomarker for early therapeutic intervention because these antibodies are associated with the development of early acute AMR, and IgM DSAs could be removed more rapidly and completely than IgG antibodies via plasma exchange. Their removal normalized the function of the transplanted grafts [122]. Another study reported that IgM DSAs protect transplanted organs from the recipient immune system during liver–kidney transplantation [123]. Conversely, there are reports suggesting that IgM DSA does not contribute to the prognosis of solid organ transplantation [7].

### 5.3. Challenging Problems

No consensus has been reached regarding the mechanism by which IgM DSA affects transplanted organs. To elucidate the clinical significance of IgM DSA, we should clarify what factors are involved, including the origin of IgM antibody-producing cells in light of the roles of IgM DSA in the context of AMR.

## 6. ABO Blood Group-Incompatible Antibodies in the Field of Transplantation

It has been reported that ABO blood group-incompatible antibodies, which consist of IgM antibodies that are naturally present and IgG antibodies that are induced upon antigen sensitization, may also influence the survival of transplanted grafts [124]. In this chapter, the involvement of ABO blood group-incompatible antibodies in transplanted organ survival, as well as the factors that affect this interference, are discussed.

### 6.1. Kidney Transplants

#### 6.1.1. Development of Immunosuppressive Therapy for ABO-Incompatible Kidney Transplants

The removal of CD20-positive B cells, or application of antidonor antibody via desensitization therapy, more effectively prevents the development of hyperacute rejection than previously established treatments. Transplant prognoses, such as graft survival and the recipient survival rate, are comparable with those of ABO-compatible-matched controls [16,125,126]. Additionally, antibody removal and conventional immunosuppression provide stable renal function for four years after transplantation, and pathological analysis revealed no indication of transplant glomerulopathy and low rates of progressive interstitial fibrosis/tubular atrophy in earlier biopsies [18]. Contrarily, other reports indicated that ABO-incompatible transplants do not require specialized immunosuppressive therapy. In one study, the survival rates of the transplanted graft and the recipient were comparable with either those in ABO-compatible-matched controls or patients with low baseline antiblood group antibody titers [14], whereas a second study recorded a graft survival rate of 100% at 36 months after transplantation [15].

#### 6.1.2. Current Outcomes of ABO-Incompatible Kidney Transplants

Although some reports indicated that the rates of early rejection and infection are high after transplantation, and the rate of graft failure is elevated early after transplantation in ABO-incompatible transplantation, other analyses reported that the recipient survival rate and long-term prognosis were equivalent to those of ABO-compatible transplantation because of the development of immunosuppression protocols [13,16,17].

### 6.2. Liver Transplants

#### 6.2.1. Development of Immunosuppressive Therapy for ABO-Incompatible Liver Transplants

Although high risk of AMR and poor prognoses is associated with ABO-incompatible liver transplant, accommodation is induced using pre and post-transplantation double-volume total plasma exchange, splenectomy, and quadruple immunosuppression (cyclophosphamide or mycophenolate mofetil, prednisone, cyclosporine or tacrolimus and OKT3 induction), and protects the transplanted graft from recipient immune attack [20].

#### 6.2.2. Current Outcomes of ABO-Incompatible Liver Transplants

In living liver transplantation, ABO-incompatible transplantation results in equivalent outcomes to ABO-compatible transplantation because of the development of immunosuppressive therapy. However, AMR remains a risk factor for graft loss in liver transplantation from brain-dead donors, as the survival and function of the transplanted graft and recipient survival were inferior because of hepatic necrosis, diffuse intrahepatic biliary complications, severe cellular rejection and late AMR [18,19,20].

### 6.3. The Role of IgM Antibodies in ABO-Incompatible Transplant Survival

It has been reported that the IgM antibody is involved in ABO-incompatible transplantation, and desensitization therapy using the IgM-type ABO antibody titer as an index is useful for preventing AMR development [127]. Contrarily, other reports stated that IgG-type ABO antibodies are risk factors for the development of AMR [128].

In addition, both IgG- and IgM-type ABO antibodies are involved in AMR development. IgG antibodies are involved in type 1 AMR, which is caused by the resensitization of IgG-producing mBCs, which are already present in secondary lymphoid tissues, by blood group antigens. By contrast, IgM antibodies are involved in type 2 AMR, which develops after initial sensitization through ABO-antigen-like substances, which are expressed on the surface of bacteria following infection early after transplantation [129]. Thus, there is disagreement regarding whether IgG or IgM ABO antibodies are involved in the development of AMR [130].

Additionally, accommodation, in which the humoral immune response to the transplanted organ is not activated despite the presence of antidonor antibodies in blood, has also been reported in ABO-incompatible transplantation. As a mechanism of accommodation induction, one possibility is that the process may be associated with a class switch from IgG- to IgG2-expressing cells. IgG2 antibodies have low ability to activate complement, and they prevent the binding of more cytotoxic isotype antibodies to grafts [131]. Conversely, exposure to a low-titer blood group antibody, and absorption of a large amount of blood group antibodies on the surface of the graft antigen, cause a series of protective changes on the surface of the graft antigen [132,133,134].

Thus, accommodation may be induced by changes in blood group antibodies or graft antigens. However, the detailed induction mechanism, the involvement of IgM antibodies, and the factors that affect the involvement are not fully understood.

### 6.4. Challenging Problems in ABO-Incompatible Transplantation

Meanwhile, high-intensity desensitization therapy has been reported to increase the risk of infections such as BK virus infection after ABO-incompatible kidney transplantation [135,136]. Therefore, methods for protecting transplanted organs from the recipient immune system using a less invasive immunosuppression protocol are urgently needed.

Meanwhile, the prognosis of transplantation differs between liver and kidney transplantation and between transplanted grafts from brain-dead and living donors in the case of liver ABO-incompatible transplantation. Therefore, it is necessary to fully understand the involvement of ABO-incompatible antibodies in transplanted organs, considering the type of transplanted organs and differences between brain-dead and living-donor transplantation, to obtain a similar prognosis between the types of grafts in ABO-incompatible transplantation. Moreover, this information will prevent severe side effects associated with high-density immunosuppression.

## 7. Significance of Complement Activation in the Field of Transplantation

C4D deposition, which is the final step of complement activation, is an AMR diagnostic criterion in the Banff classification for organ transplants. In cases of C1q-positive DSA, the development of AMR and incidence of glomerulopathy are significantly more common, and the survival rate of the transplanted graft is significantly lower than observed in C1q-negative cases [137,138,139]. Additionally, the effectiveness of anticomplement drugs for AMR treatment has also been reported. The administration of eculizumab, a humanized anti-C5 monoclonal antibody against active AMR, in the early stage after transplantation generally improved the pathological findings and function of transplanted grafts [140]. Eculizumab administration was also effective in preventing AMR development after transplantation, even in pre-DSA–positive cases of living kidney transplantation [141].

Based on these data, the therapeutic effect of anticomplement drugs on the progression of pathological conditions in AMR has been indicated. Therefore, complement activity is closely related to tissue damage in AMR, and the early detection of complement activity and therapeutic intervention may be useful for preventing and treating AMR [142].

## 8. Clinical Significance of IgM-Based Therapies in the Field of Transplantation

In the field of transplantation, improvement of the prognosis of organ transplant recipients has been indicated by a significantly higher DSA clearance rate in patients treated with IgM antibody-enriched IVIG in the case of de novo DSA production after lung transplantation [143]. DSA was cleared in 91% of patients after three months on average, and the DSA level was decreased in 12% of patients by IgM antibody-enriched IVIG. The rates of biopsy-confirmed rejection and transplanted graft survival rate were not significantly different from those in the control group (de novo DSA-negative) [35].

From these findings, it was possible to infer the mechanism by which IgM antibodies of different origins participate in the development of AMR. Using these characteristics of the IgM antibodies and IgM receptor, their possible clinical application in AMR control and their drawbacks are discussed in the next chapter.

## 9. The Possibilities of IgM Antibody Manufacturing for AMR Control

### 9.1. B-1 Cell-Derived Natural IgM Antibodies

It is described in this review that IgM antibodies produced by each B cell subset have different mechanisms of engaging with target organs. Furthermore, a comprehensive summary that indicates the possible clinical application of these IgM antibodies is provided, including the risks that need attention. Intravenous B-1 cell-derived natural IgM antibodies also seem promising in controlling damage in organs in patients with autoimmune diseases and organ transplant recipients.

Some problems requiring attention include the difficulty in differentiating B-1 cells into antibody-producing cells in vitro because self-antigens and environmental factors play important roles in the survival and growth of such cells in vivo [30,31]. Therefore, clinical strategies to maintain the supply of B-1 cell-derived IgM antibodies are urgently required. Moreover, if it is possible to explore the involvement of natural IgM antibodies in target organs by classifying them into B-1a and B-1b cell-derived antibodies, it may contribute significantly to the development of more effective therapeutic methods.

### 9.2. B-2 Cell-Derived IgM Antibody

The detection of DSA-specific IgM mBC differentiation may be useful in evaluating pathological conditions and reaching an early diagnosis for AMR control. Although the differentiation of B-2 cells into antibody-producing cells may occur in vitro, and antibody levels in the culture supernatant can be analyzed, the culture period might cause a delay in definitive diagnosis. Therefore, it may be useful to develop a diagnostic technique capable of selectively detecting DSA-specific IgM mBC differentiation instantly as a solution.

In addition, if it is possible to clarify the mechanism of the involvement of IgM antibodies produced by B-2-derived antibody-producing cells in transplanted grafts, a more detailed method for evaluating the pathology of AMR may be established. Therefore, diagnostic technology capable of selectively analyzing B-2 cell derived IgM antibodies is needed.

### 9.3. FcµR

Regulating signals from FcµR may maintain the balance of humoral immune reaction against the target organ referring to the involvement of IgM receptor in pathological conditions. Therefore, if a structural or functional abnormality of FcµR is detected, it may be possible to apply it as a biomarker in the development of AMR. Furthermore, infusion of the extracellular domain of the IgM receptor (FcµR) or anti-FcµR monoclonal antibody may effectively protect the target organs from recipient immune attack. Further elucidation of the mechanism by which the IgM antibody participates in humoral immunity against target organs via FcµR is required for its therapeutic application in the context of AMR.

### 9.4. Current Status of the Clinical Application of IgM Antibodies of Different Origins as a Standard Diagnostic and Therapeutic Entity

We have discussed techniques that can be used for the production, purification and quality control of IgM antibodies from each cell origin, as well as the problems to be solved for their clinical application as a standard-of-care procedure or a diagnostic tool in the context of AMR.

First, with regard to the clinical application of IgM antibodies as a therapeutic strategy, we particularly examined the possibility of protection of the target organ from the humoral immune reaction of B-1 cell-derived IgM antibodies. Surface markers of human-derived B-1 cells have also been clarified [27,28] and can be used to separate specific cell subtypes through fluorescence-activated cell sorting [144]. Therefore, if an in vitro assay system capable of inducing B-cell differentiation into PCs can be established, it would be possible to collect the IgM antibodies derived from B-1 cells. Alternatively, it has been reported that serum polyclonal antibodies can be specifically separated using antigen-immunoaffinity chromatography [145,146].

However, since the ratio of B-1 cells in human PBMCs is low [8], a large amount of donated blood is required for the effective therapeutic application of IgM-type polyclonal antibodies derived from B-1 cells. Therefore, methods such as the production of specific antibody-producing B-1-cell hybridomas are required for ensuring a stable supply of B-1 cell-derived IgM antibodies. Currently, however, human-derived antibody-producing cell hybridomas may be difficult to produce.

Next, the possible clinical application of B-2 cell-derived IgM antibodies for the pathological evaluation of AMR was discussed. LABScreen Single Antigen (One Lambda) has been used to identify the antigen specificity of serum HLA antibodies. Additionally, the IgM class of HLA antibodies can be analyzed using IgM antibody as the secondary antibody. Likewise, the IgG subclass of HLA antibodies can be detected using the secondary antibody for each subclass [147,148]. However, most of the serum IgM antibodies are natural antibodies derived from B-1 cells, and monoclonal IgM antibodies derived from B-2 cells need to be detected selectively. It has been reported that it is possible to analyze the antigen specificity of IgM^+^ mBCs circulating in the periphery through use of the in vitro PBMC assay system, without interference from the natural antibodies [94].

It was also reported that B cells expressing monoclonal antibody (mAb) on the cell surface could be detected at the single-cell level with a high probability. This was achieved by passing the cells through antigen-conjugated fluorescent beads, whereupon the cell surface mAb would bind to the target antigen via the antigen–antibody reaction [149].

Although it is possible to detect IgM^+^ B-2 cells corresponding to specific antigens, further clarification of the conditions for inducing the differentiation of these cells into PCs is needed, especially in vivo, because it has been reported that the induction of mBCs differentiation into PCs may be suppressed by the effects of accessory cells including natural killer cells and monocytes in vivo [150,151].

Therefore, there are many issues to be resolved before IgM antibodies of different origins can be applied clinically as a standard-of-care or diagnostic procedure.

## 10. Concluding Remarks

The clinical significance of IgM antibodies has been evaluated in fields such as infection, autoimmune diseases and cancers, and their potential organ-protective and anti-inflammatory effects, as well as utility in evaluating pathological conditions, have been reported. However, the clinical significance of IgM antibodies has not received substantial attention because no consensus has yet been reached regarding the mechanism by which IgM antibodies participate in the survival of transplanted grafts. Such an inconsistency in the role of IgM antibodies in the context of AMR may arise from the differences in the origin of IgM antibody-producing cells and their target organs, and other conditions.

Many reports indicate that IgM antibodies protect transplanted grafts from immune attack via mechanisms such as (1) the induction of anti-inflammatory activity, (2) suppression of complement activation, (3) IgG autoantibody production. The B-1 cell-derived IgM antibody may be involved in such protective effects considering the mechanisms by which IgM antibodies of different origins participate in the development of various diseases.

Regarding B-2 cell-derived IgM^+^ mBCs, some studies revealed that the detection of DSA-specific IgM^+^ mBCs is useful for evaluating pathological conditions and facilitating early diagnoses in the context of AMR because some of these cells undergo class-switching into IgG^+^ mBCs upon activation of the humoral immune response against the transplanted grafts.

In addition, the detection of B-2 cell-derived IgM antibodies corresponding to donor-specific HLA may not have been thoroughly evaluated because of the presence of B-1 cell-derived IgM antibodies in serum. Thus selective analysis of these antibodies may lead to a more detailed understanding of the pathophysiology of AMR.

In conclusion, understanding the direct or indirect involvement of IgM antibodies in the activation of humoral immunity against transplanted grafts, by considering the original corresponding antigen, differences in the type of target organs as well as other conditions that are yet to be defined, may contribute to the control of AMR, which will make it possible to conduct transplantation with favorable long-term outcomes in the future. We should develop and make full use of modern production, purification, and quality control techniques to realize the clinical application of the organ protection effect of B-1 cell-derived IgM antibodies, and their usefulness as a method for evaluating the humoral immune activation of B-2 cell-derived IgM antibodies, before IgM antibodies can be recognized as main players in diagnosis and therapy in the context of AMR. (Figure 3).

## Figures and Tables

**Figure 1 pathogens-10-00004-f001:**
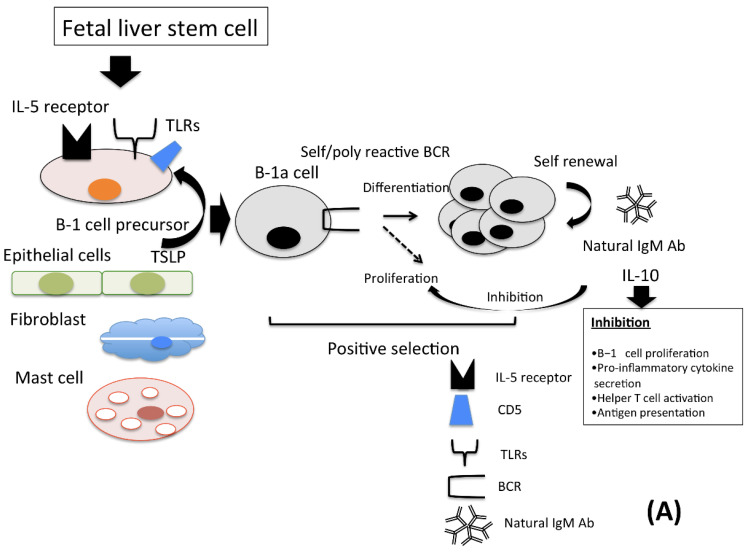
Despite the environmental factors and thymic stromal lymphopoietin (TSLP) and signals from IL-5 receptors, TLRs plays an important role in prolonging B-1 cell survival, and those cells expressing BCR with affinity for antigen are Scheme 1. Cells are classified into two types, namely, B-1a and B-1b. (**A**). The B-1a cell precursor develops from the liver fetal stem cell, and B-1a cell-derived antibody-producing cells produce natural IgM antibodies, which react to environmental antigens in a polyclonal manner, and anti-inflammatory cytokine IL-10. (**B**). The B-1b cell precursor develops from the liver fetal stem cell during the fetal period, and from bone marrow in adult. B-1b cell-derived antibody-producing cells produce antigen specific IgM and IgG3, IgA anTable 10. Tool-like receptors, TLRs; thymic stromal lymphopoietin, TSLP; T cell independent antigen, B cell receptor, BCR; T cell independent, TI; antibody, Ab.

**Figure 2 pathogens-10-00004-f002:**
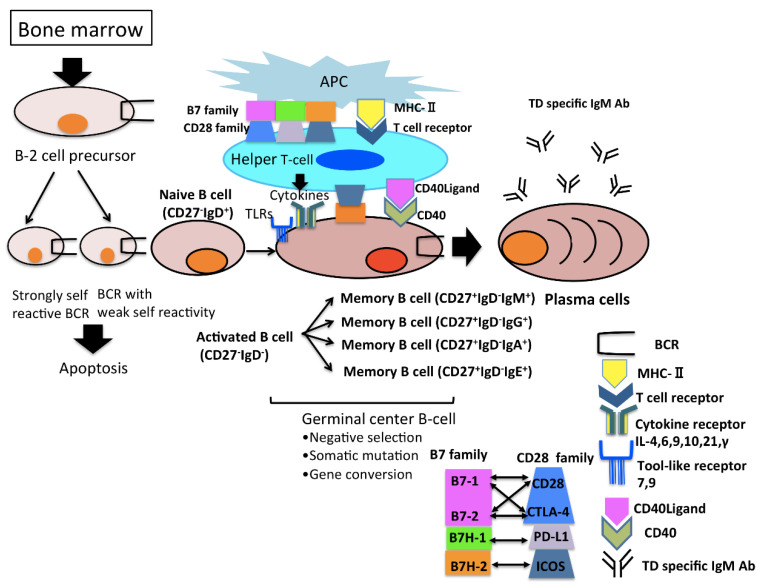
B-2 cells originate from bone marrow, yet B-2 cells with a low affinity for foreign antigens or a high affinity for self-antigens are excluded. BCR; B cell receptor, APC; antigen presenting cell, ICOS; inducible T-cell co-stimulator, PD-1; programmed cell death protein-1, CTLA4; cytotoxic T-lymphocyte associated antigen 4, MHC; major histocompatibility antigen complex, TCR; T cell receptor, TD; T cell dependent, Ab; antibody.

**Figure 3 pathogens-10-00004-f003:**
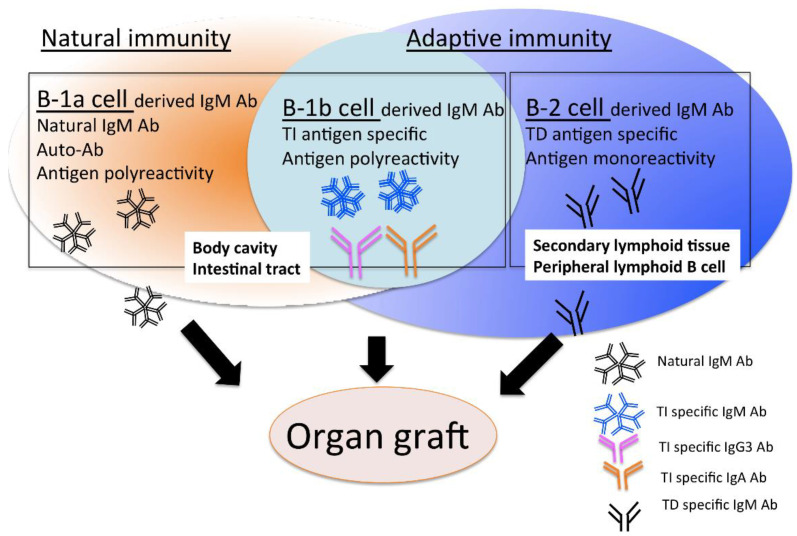
Most serum IgM antibodies comprise natural antibodies with low antigen affinity derived from B-1 cells (B-1a cells and B1-b cells), while some of these antibodies may contain IgM antibodies derived from B-2 cells (conventional B cells) with a high affinity to the antigen expressed on the graft. It is crucial for the control of AMR to elucidate the characteristics of IgM antibodies and the mechanism in which the IgM antibodies are involved in allograft rejection, with consideration of differences in the origins of antibody-producing cells. TI; T cell dependent antigen, TD; antibody, Ab. TI; T cell dependent antigen, TD; antibody, Ab.

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
