# Peer review of "Characteristics of Immunoglobulin M Type Antibodies of Different Origins from the Immunologic and Clinical Viewpoints and Their Application in Controlling Antibody-Mediated Allograft Rejection"

_pathogens, 2020, doi:10.3390/pathogens10010004_

Round 1

Reviewer 1 Report

Dear Authors,

the manuscript has been improved through the revision. I am still convinced that the knowledge on IgM role in various therapeutic settings, especially in transplantation, is valuable to the scientific community as well as to contemporary healthcare. However, my main point has remained unaddressed: the manufacturing and quality control of IgM antibodies, required for their detection and development, to lead to the recognition of this class of molecules as main players in diagnosis and therapy. Your comments in point 9 are valuable, but can be summarized as conclusion remarks; what the manuscript truely misses is the evaluation of techniques rendering IgM a standard diagnostic and therapeutic entity, including modern production, purification and quality control techniques. Please consider the content in novel references, which supports your ideas and will help to propell IgM as a marker as well as a therapeutic entity into the midpoint of standard-of-care procedures, which I am also sure is the main purpose of your review.

Please find below a list of minor remarks that should be addressed, and above all please consider the minor changes in the Figures.

Line 35: complement activation

Line 171: the reference to this important finding is missing

Line 175: P. acnes in italics, Gram in caps

Line 180-181, Figure 1: IgG3 is still a pentamer

Line 191: different font and size

Line 196: Toll-like receptors

Lines 196-197: these lines appear to be the abbreviations pertaining to Figure 1, and seem to be misplaced?

Line 215: it has been reported

Line 215: CpG-ODN, abbreviation is missing

Line 233: IgM(positive) IgD(negative), please correct the capitalization (remove the superscript from I)

Line 251: suppresses T-cell activation

Line 253: and enhances secretion

Lines 258-261: I think these should be abbreviations pertaining to Figure 2?

Line 352-356:

Line 376: CD81 or TAPA-1 is a suitable abbreviation

Line 376: in lowering the threshold/ or in a lower threshold

Line 468: IgM-enriched IVIG effectively suppresses

Line 478: evaluating pathological conditions

Lines 743-783: This paragraph misses the point; the contents are valuable but can be included in concluding remarks section. Please consider commenting on novel methods for production, purificarion and validation of IgMs intended for therapy and diagnosis instead.

Reviewer 2 Report

In the revised manuscript by Matsuda et al. they provide a broad overview about antibodies in allograft rejection, with of focus on IgM antibodies here. They cover our current understanding of the pathophysiology, the involvement of B-1 and -2 cells and IgG/IgM antibodies in this process. They substantially changed the manuscript in the revision and therefore increased the quality of the review. I have no further comments.

Author Response

We really appreciate your meaningful suggenstions.

Reviewer 3 Report

The authors have addressed my comments

Author Response

We really appreciate your meaningful suggestions.

This manuscript is a resubmission of an earlier submission. The following is a list of the peer review reports and author responses from that submission.

Round 1

Reviewer 1 Report

An interesting review on the role of IgM in transplantation.

Some comments:

1) Could the outcomes of HLAi kidney transplantation be more in depth covered with a specific section as for the ABOi?

2) The authors could explore the mechanisms why tolerance is more common for liver rather than kidney transplantation, highlighting what could be achieved in the latter in views of the possibilities in common among the two fields

Reviewer 2 Report

The review provided by Matsuda et al. summarizes the current knowledge on IgM antibodies in allograft rejection. They first gave insights in the pathophysiology and then discussed the clinical role of IgM antibodies in organ transplantation. The manuscript is clearly written also for people outside the field. The authors also have expertise in this field as shown by recent publications. In my opinion, two issues might be addressed:

  • References: Reference 1, 14 and 36, as well as reference 22 and 44 are the same. Please carefully check again the references.
  • Figures: A short description in more detail on what is shown in the figure would be helpful.
  • Figure 1: Dividing the figure in A and B is in my view also helpful.

Reviewer 3 Report

In this very systematic and comprehensive review, the novel insights into the funtionality of IgM antibodies and especially their potential significance for prognosis and therapeutic application in transplantations are well covered. The article encompasses recent discoveries on this class of antibodies on the molecular level, cellular level (regarding cell and type of producing cell) as well as their emerging role in diverse clinical settings. With this, my only major recommendation would be to include a short paragraph on the possibilities of their manufacturing for biopharmaceutical production, as this is a required step before their broader use for diagnostics and therapy. Please find below a list of minor comments, which I however feel should be addressed for the clarity and partially the structure of the manuscript to be improved.

Line 19: „uncertainty“ is a too broad expression, especially for abstract. Would you rephrase this to uncertain specificity of ambiguous specificity or similar?

Line 59: antigen affinity is low (please cite the typical affinities)

Line 80: quotation in brackets is in an unusual format

Line 102: without support

Line 103: Figure 1 (as well as all other figures) has an unnecessary label Figure.1. The elements of  the figure require an explanation in the legend. Please review the elements on the right hand-side of the Figure as they are not self-explanatory.

Line 116: Please improve the labels in Figure 2, various members of the B7 family are only labeled „B7“ – please differentiate according to the interacting molecule.

Line 119:  PD-1 is programmed cell death protein -1

Line 121, Table 1: Immunoglobulin produced; Antigen affinity and specificity; Adaptive immune response; Pentamer formation and complement activation

Line 128: cross-link the BCR

Line 129: by trapping ICs in follicular…

Line 142: (Fab‘)2, 2 in subscript

Line 145: … promoted the deposition… and led to the deterioration

Line 150: followed by the induction of an inflammatory response

Line 150: citation is missing

Line 152: the activity of these antibodies in the target tissue, taking into account their different origin

Line 156: anti-inflammatory activity in target organs, citation is missing

Line 164: complement activation

Line 183: Toll-like receptor, please also introduce the abbreviation TLR as it is used in the line 238

Line 184: PBMCs

Line 192: of B-1 and B-2 cells

Line 196: polymeric?

Line 203: to lower the threshold

Line 241: …in immune response

Line 242: „ Significance of IgM antibodies in various disease states“ would be a better title

Line 274: „Importance of IgM-based therapies“  or „Clinical applications of IgM based therapies“ would be sound better. I would recommend to move the paragraph on IgM in transplantation to the dedicated section 9 and in this case rephrase the title to: Clinical significance of IgM-based therapies in inflammation and infection.

Line 282: …together with erythropoietin.

Line 347-349: this sentence is not clear. Does it mean: with the development of immunosuppression, such as after a combination of monoclonal etc., it becomes possible to prevent…?

Line 349: hyperacute rejection

Line 428: the role of IgM antibodies involved in…

Line 438: References are in an unusual format, shouldn’t DOI be included?
